# Diagnostic Value of Radio-Guided Sentinel Node Detection in Patients with Prostate Cancer Undergoing Radical Prostatectomy with Modified-Extended Lymphadenectomy

**DOI:** 10.3390/cancers14205012

**Published:** 2022-10-13

**Authors:** Bartosz Małkiewicz, Błażej Bugla, Maciej Czarnecki, Jakub Karwacki, Paulina Długosz, Adam Gurwin, Paweł Kiełb, Artur Lemiński, Wojciech Krajewski, Diana Jędrzejuk, Marek Bolanowski, Agnieszka Hałoń, Tomasz Szydełko

**Affiliations:** 1University Center of Excellence in Urology, Department of Minimally Invasive and Robotic Urology, Wroclaw Medical University, 50-556 Wroclaw, Poland; 2Department of Urology and Urological Oncology, Pomeranian Medical University, Powstańców Wielkopolskich 72, 70-111 Szczecin, Poland; 3Department of Endocrinology, Diabetes and Isotope Therapy, Wroclaw Medical University, 50-367 Wrocław, Poland; 4Department of Clinical and Experimental Pathology, Wrocław Medical University, 50-556 Wrocław, Poland

**Keywords:** prostate cancer, radio-guided lymph node dissection, radical prostatectomy, pelvic lymph node dissection, sentinel lymph node dissection

## Abstract

**Simple Summary:**

Radio-guided sentinel lymph node (SLN) detection techniques are utilized in many malignancies, including breast and penile cancer. In the setting of prostate cancer (PCa), SLNs identification had low specificity and sensitivity in available studies, and extended pelvic lymph node dissection (ePLND) remains the gold standard in the assessment of nodal invasion. Our study is the first to detect SLNs pre- and intraoperatively while performing ‘whole-pelvis’ modified-extended lymphadenectomy (mePLND), including lymph nodes (LNs) from the ePLND template, as well as the presacral and Marcille’s fossa areas. The aim of this study was to assess the efficacy of single-photon emission computed tomography (SPECT-CT) and the gamma-probe in the SLN detection, as well as to map lymphatic drainage patterns in the prostate. Results showed that the SLN detection technique has very low specificity and relatively high sensitivity and remains a poor nodal staging tool in intermediate- and high-risk PCa patients.

**Abstract:**

**Background.** In many malignancies, sentinel lymph node dissection (SLND) is being used as a nodal staging tool. We prospectively evaluated the diagnostic value of radio-guided sentinel lymph node (SLN) detection in patients with prostate cancer (PCa). This study aimed to investigate the reliability of the radio-guided SLN detection technique for perioperative localization of LNs metastases as well as to map lymphatic drainage patterns of the prostate. **Methods.** Forty-three patients with intermediate- or high-risk cN0cM0 PCa at conventional imaging underwent radical prostatectomy with modified-extended pelvic lymph node dissection (mePLND). A day before the planned surgery, a Tc-99m nanocolloid was injected into the prostate under the control of transrectal ultrasonography (TRUS). Preoperative single-photon emission computed tomography (SPECT-CT) imaging and intraoperative gamma-probe were used to identify SLNs. All positive lesions were excised, followed by mePLND. The excised lymph nodes (LNs) were then submitted for histopathological examination, which was used as a reference for the calculation of diagnostic parameters of the SLN technique for SPECT-CT and the intraoperative gamma-probe. **Results.** In total, 119 SLNs were detected preoperatively (SPECT-CT) and 118 intraoperatively (gamma-probe). The study revealed that both SLN detection techniques showed a sensitivity of 90% and a specificity of 6.06%. The negative predictive value (NPV) was 66.67%. SLN technique would have correctly staged nine of 10 patients, which is the same result as in the case of limited LND. However, it allowed the removal of all metastatic nodes only in four of them. SLND would have comprised 69.7% of preoperatively detected LNs, and removed 13 out of 19 positive LNs (68.42%), respectively. **Conclusions.** Radio-guided SLND has a low diagnostic rate and is a poor staging tool. ePLND remains the gold standard in nodal metastases assessment in PCa. Our study indicates that lymphatic drainage of the prostate and actual metastasis routes may vary significantly.

## 1. Introduction

Prostate cancer (PCa) is the second most frequent cancer and the fifth leading cause of cancer-related death among men worldwide. According to statistics, in 2020, it was the most frequently diagnosed cancer in men in over half of all the countries of the world, with 1,414,259 new cases [1]. As the presence of lymph node (LN) metastases is a major unfavorable factor for recurrence and survival in men with PCa, their accurate identification is crucial, especially in choosing an adequate adjuvant therapy [2,3].

Computed tomography (CT) and magnetic resonance imaging (MRI) is the most common radiologic imaging techniques assessing the shape and size of LNs. Despite engaging novel imaging modalities, e.g., prostate-specific membrane antigen (PSMA) positron emission tomography (PET) or single-photon emission computed tomography (SPECT-CT), the nodal staging accuracy of radiologic imaging techniques remains suboptimal [4,5].

Thus, extended pelvic lymph node dissection (ePLND) remains the most optimal form of nodal invasion assessment. Lymphadenectomy, aimed primarily at evaluating nodal metastasis presence in PCa patients, could potentially have a therapeutic value as well. Unfortunately, some recent studies have found that men with PCa who received ePLND did not have more favorable oncological outcomes than those who received limited PLND (lPLND) [6,7]. Furthermore, PLND, like any surgical intervention, carries a risk of complications, both peri- and postoperative. It correlates with the extent of the operation—the more extensive the template, the greater the chance of complications [8].

Another technique used for assessing lymph node involvement is sentinel lymph node dissection (SLND). In many other cancers, such as breast cancer, melanoma or penile cancer, SLND has already appeared in the guidelines and is a very effective nodal staging tool [9]. In the setting of PCa, SLND has been thoroughly tested in recent years and has a status of an experimental but promising staging tool. Moreover, the study by Grivas et al. confirmed that adding SLND to ePLND improves BCR-free outcomes compared with ePLND only [9]. According to some researchers, like Egawa et al., a patient is more likely LN-negative if he is sentinel lymph node (SLN) metastasis-free [10]. SLND is an attempt to find the balance between the low complication burden of SLND and the high staging accuracy of ePLND.

To secure the complete excision of all prostatic lymphatic drainage areas, we have extended the template of PLND with additional LNs stations. Therefore, the surgical template of PLND used in our study should be described as ‘modified-extended’ (mePLND) or ‘whole-pelvis,’ as it includes the presacral area and the fossa of Marcille, an anatomical region limited by the fifth lumbar vertebra medially, the inner edge of the large psoas muscle laterally, and the upper edge of the wing of the sacrum caudally [11]. LNs of these regions are connected to the prostate lymphatic system [12].

The primary aim of this study was to assess the efficacy of perioperative SLN detection techniques (SPECT-CT and the gamma-probe) in PCa patients selected from the European Association of Urology (EAU) intermediate- and high-risk groups verified by PLND template covering the whole pelvis. We also compared lymphatic drainage patterns with actual metastatic routes, investigating the mapping aspect of SLNs in PCa. The current study aims at establishing the benefits and harms of this approach in the management of PCa and the search for future possibilities.

## 2. Patients and Methods

### 2.1. Patients and Study Design

The study included patients with localized or locally advanced prostate cancer (cT1-3bN0M0) indicated for RP with PLND who met the following criteria: age ≥18 yr, written consent to participate in the study, histologically proven intermediate- or high-risk prostate cancer according to the EAU risk groups. The exclusion criteria included previous irradiation or pelvic surgery, American Society of Anesthesiology classification >3, any coagulation disorder, refusing pelvic CT scans, pelvic MRI or bone scintigraphy, and existing contraindications for performing PLND. The study was approved by the Regional Ethics Committee (KB-568/2017), and written informed consent was obtained from each patient.

### 2.2. Radiocolloid Injection and SPECT-CT Imaging

Injections of Tc-99m nanocolloid (Nanocoll; GE Healthcare, Milano, Italy) were applied during TRUS and under local anesthesia performed one day before planned surgery. One radionuclide injection of 50 MBq (1.0 mL) was applied per lobe using a Chiba needle (0.95 220 mm). Injections were applied bilaterally in the line between transitional and peripheral zones of the prostate. Due to the high frequency of multifocal PCa and the difficulty in visualizing cancer foci in TRUS, the SLN procedure differs from other tumors, and the radiotracer is not administered peri/intratumorally. After 2–3 h, hybrid SPECT-CT lymphoscintigraphy was performed with the use of Gamma Camera BrightView XCT (Philips Healthcare, San Jose, CA, USA) equipped with low-energy high-resolution collimators (LEHR). The low-dose CT projection was applied without contrast. The study time was approximately 30 min. The following image acquisition parameters were used: matrix size 64 × 64, 64 strokes for each head, counting time 20 s per image and 3 fields, 1 mm layer, 512 × 512 matrix size, 120 keV voltage, 20 mA current, standard and iterative reconstruction, 120 keV voltage, 20 mA current for SPECT and XCT projections respectively. The SOFT TISSUE filter (0.6) and the option for breathing correction were used. In order to facilitate the intraoperative anatomical location of active foci, images obtained from SPECT and CT were fused and reconstructed using the Extended Brilliance Workspace V1.0 software (Philips Medical Systems Nederland B.V., Eindhoven, The Netherlands). Radionuclide uptake sites whose activity was significantly higher than background and topographically unrelated to the injection site, rectum, bone marrow, kidneys or liver were considered draining SLNs. The reconstructed images were then used for intraoperative navigation and localizing of SLNs (Figure 1).

### 2.3. Surgical Procedure

All procedures were performed through a retroperitoneal approach. First, under the control of the SPECT-CT fusion images (Figure 1), all lymph drainage stations were methodically scanned using the FlexProbe CCXS-OP-FP handheld gamma radiation probe (Crystal Photonics Gmbh, Berlin, Germany). A probe with a collimator and a head viewing angle was used, allowing precise orientation at the test site and minimizing the risk of artifacts. A constant, intensive radioactivity reading being at least 2 times the background (fatty tissue of each patient) measurement was considered a positive signal. After locating the SLNs, tissues were selectively removed and mapped on a diagram (Figure 2a). Then, all patients underwent bilateral modified-extended lymphadenectomy and re-mapped the scheme (Figure 2b). The upper limit of the mePLND was at the level of the aortic bifurcation. The caudal extension of LND was at the level of Cooper’s ligament, with the genitofemoral nerve as the lateral boundary. Lymphatic and connective tissue were removed from both sides of the pelvis separately from the following anatomical sites: obturator, external iliac, internal iliac, presacral, Marcille’s fossa and common iliac. The removed lymph and fibrous-fatty tissues from each anatomical area of the lymphadenectomy were screened ex vivo for the presence of hot-spot foci that could be missed at the first exposure and then mapped on the template. The excised lymph nodes were then submitted for histopathological examination as separate specimens. In the last stage, an open radical prostatectomy was performed.

### 2.4. Definitions and Statistical Analysis

Data are described with mean, median and range for continuous variables and rates (as well as percentages) for latent variables. SLN was defined as a lymph node expressing radioactivity at each anatomical LN station. The compatibility between the SLN pathological evaluation and definitive pathological status of LNs in the same anatomical site was calculated by the SLN detection in PCa surgery diagnostic tests of sensitivity, specificity, accuracy, false negative rate and positive and negative predictive values. The test results were defined as follows: true positive (TP)—SLNs detected, and metastasis found in at least one SLN; false positive (FP)—SLNs detected, but no metastases in SLNs; true negative (TN)—no SLNs detected and no metastases in SLNs and other LNs; and false negative (FN)—SLNs detected, but metastases found only in nodes other than SLNs. The individual parameters of the diagnostic test were calculated on the basis of the following formulas: sensitivity = (TP/(TP + FN)) × 100, specificity = (TN/(FP + TN)) × 100, positive predictive value PPV = (TP/(TP + FP)) × 100, negative predictive value NPV = (TN/(TN + FN)) × 100, ACC accuracy = ((TP + TN)/TP + FP + TN + FN) × 100, FN rate = (FN/(TP + FN)) × 100. Assuming that pathological examination used as the reference method gives 100% certainty in determining the presence of metastases and that both SLN detection methods using gamma-probe and SPECT-CT have undetermined diagnostic values. Thus, the study group was divided into 4 parts. All statistical analyses were performed using Statistica v.13.3 (TIBCO Software Inc., Palo Alto, CA, USA). 

The groups of LNs included in each type of PLND differ depending on the study. In our work, we classified lymph nodes into particular templates based on the experience of our site. The definition of specific types of PLND in our study differs from the one that can be found in the guidelines of the EAU. Limited PLND involved obturator and external iliac regions. Standard lymphadenectomy included internal iliac and external iliac regions, as well as the obturator fossa. Extended PLND included obturator, external, internal, and common iliac, while the modified-extended template was defined as extended with the addition of the presacral and Marcille’s fossa regions. Figure 3 shows the scope of mePLND.

## 3. Results

### 3.1. Basic Clinicopathological Characteristics

Between September 2017 and August 2020, the SLN detection technique and RP with mePLND were performed on 43 patients. Table 1 depicts the characteristics of patients included in this analysis. The median age and PSA at surgery were 63.4 yrs and 14.02 ng/mL, respectively. Twenty-five patients (58%) had non-organ-confined tumors (pT3a-pT3b), and all patients had negative surgical margins. The pathological grade groups were 1, 2, 3, 4, and 5 in four (9.4%), 21 (48.7%), 13 (30.2%), one (2.3%), and four (9.4%) patients, respectively. The median risk of LNI, according to the Briganti nomogram, was 9%. The mean number of harvested LNs was 26 (range: 14–52) per patient. LN metastasis was found in ten (23%) of the patients, of whom seven (16%) had single LN metastasis (N1 stage), and three (7%) had multiple LN metastases (N2 stage).

### 3.2. Sentinel Nodes Detected Scintigraphically

In 2 of 43 patients (4.7%), no SLNs were detected using preoperative SPECT-CT. In total, 119 SLNs were detected (median: 2; interquartile range [IQR]: 2–4), and 115 of them were identified and assigned to specific LN regions: obturator (n = 44; 37%), external iliac (n = 32; 26.9%), internal iliac (n = 7; 5.9%), presacral (n = 17; 14.3%), Marcille’s fossa (n = 5; 4.2%), and common iliac (n = 10; 8.4%). In four of 43 patients (9.3%), the quantity of LNs detected by SPECT-CT and the gamma-probe differed from each other. Thus, four LNs (3.4%) couldn’t be distributed with certainty before the intraoperative detection. In three of the above-mentioned patients, SPECT-CT revealed more SLNs; in 1 patient, the gamma-probe detected more LNs than preoperative scintigraphy. In 21 patients (48.8%), excluding patients whose pre- and intraoperative results varied, SLNs were distributed unilaterally. The revealed lymph drainage pattern indicates that the SLND template comprises 69.7% of preoperatively detected LNs, while ePLND enfolds 84% of them. Overall, 13 LNs detected with SPECT-CT occurred metastatic. During the histopathological examination, LNI was confirmed in 19 LNs; thus, SPECT-CT presented only 68.4% of all positive LNs.

### 3.3. Sentinel Nodes Detected Intraoperatively

In total, 118 LNs were revealed intraoperatively using the gamma-probe (median: 2; IQR: 2–4) and distributed as follows: obturator (n = 44; 37.3%), iliac external (n = 33; 28%), iliac internal (n = 7; 5.9%), presacral (n = 18; 15.3%), Marcille’s fossa (n = 5; 4.2%), and common iliac (n = 11; 9.3%). Among 118 hot-spots, 13 occurred invaded, and were distributed as follows: obturator fossa (n = 5; 38.46%), external iliac (n = 7; 53.85%), and internal iliac (n = 1; 7.69%). Metastatic LNs revealed pre- and intraoperatively were detected in nine patients. 

### 3.4. Diagnostic Parameters of the SLN Technique for SPECT-CT and the Gamma-Probe

Based on the obtained pathological data and the results of the SLN detection, the diagnostic parameters of the SLN technique for SPECT-CT and the intraoperative gamma-probe were calculated. The structure of the results in the analyzed group was as follows: TP-9, FP-31, TN-2 and FN-1. Diagnostic values for sensitivity, specificity, positive predictive value, and false-negative rate for the SLN gamma-probe technique were 90.00%, 6.06%, 22.50% and 10.00%, respectively. The remaining diagnostic parameters of the gamma-probe and SPECT-CT are presented in Table 2. 

### 3.5. Lymphadenectomy and Positive Lymph Nodes

The bilateral mePLND combined with the SLN procedure resulted in 1097 LNs (median: 26; IQR: 14–52) removed in 43 patients. Most LNs were removed from the obturator fossa region (474/1097 = 43.26%) and from the external iliac region (302/1097 = 27.52%). Other areas were the Marcille’s fossa region (107/1097 = 9.74%), the common iliac region (101/1097 = 9.21%), the presacral region (84/1097 = 7.65%), and the internal iliac region (29/1097 = 2.62%). A total of 19 LN+ (median: 1; IQR:1–3) were found in 10 of 43 patients (23.26%). The predominant site for LN+ was the obturator fossa region (n = 9; 47.37%), followed by the external iliac region (n = 8; 42.11%) and the interior iliac region (n = 2; 10.53%). The metastases were located only in these three regions. In other areas, no metastases were found (Figure 4). 

The lPLND (external iliac and obturator fossa regions) would have correctly staged nine of 10 patients (90%) and would have removed all LN+ in eight of them (80%). A standard LND would have correctly staged all patients (100%) and additionally would have removed all LN+ in all patients. Removing only SLNs would have correctly staged nine of 10 patients, which is the same result as in the case of lPLND. However, only four of them would have all metastases removed. Table 3 shows the staging accuracy of different variants of LND. Table 4 shows the characteristics of the removed SLNs specified for each anatomical region.

### 3.6. Morbidity

Postoperative morbidity connected with lymphadenectomy was recognized in 13 out of 43 patients (30.23%). Among the complications associated directly with LND, there was lymphorrhea, which occurred in seven patients, and a symptomatic lymphocele in one patient.

## 4. Discussion

The results of our study indicate that the radio-guided SLN detection, based on preoperative SPECT-CT and intraoperative usage of the gamma-probe, is a poor diagnostic method in terms of assessing LNI in PCa. Both techniques showed a sensitivity of 90% but a specificity of only 6.06%. The SLN detection procedures showed an accuracy of 25.58%. The mapping aspect of our research confirms a significant heterogeneity in the lymphatic drainage pattern characteristic of a prostate. These findings strongly indicate that SLN detection tools are not yet sufficient to be used as staging tools in a clinical setting and require further research.

Although non-invasive methods of nodal staging in PCa are developing rapidly, lymphadenectomy remains a gold standard nodal staging tool, which is vital, especially in patients at intermediate and high risk of nodal involvement [13]. Nevertheless, the proper LND template remains an unresolved issue and requires thorough investigation, as the adverse effects burden increases with the quantity of LNs removed [14]. 

Establishing the precise localization of LN metastases allowed us to compare different templates of LND. According to the EAU PCa Guidelines, only extended and super-extended PLND templates comprise internal iliac LNs [15]. However, our study shows that the ratio of the number of excised LN+ to the total number of removed LNs in this region is relatively high (25%, 1 of 4 LNs), which suggests that this might be the next landing site for metastases following obturator and external iliac nodes. There are other studies leading to a similar conclusion. Therefore, it should not be left out during any lymphadenectomy [16,17,18]. In our study, SLND would correctly stage 90% of patients, although only 40% would have all of the metastatic LNs removed. In addition, it would cover only 68.4% of metastatic LNs. On the other hand, the standard LND template (described as extended in the EAU guidelines and comprising LNs of the obturator fossa and the external and internal iliac areas) would cover all of the involved LNs. 

It is important to mention that despite the usage of novel imaging modalities such as prostate-specific membrane antigen (PSMA), positron emission tomography (PET), or PSMA/MRI, it remains unreachable to detect SLNs and stage patients with efficacy. Gandaglia et al. investigated PSMA radio-guided surgery in the setting of robot-assisted RP. The detection technique showed 63% sensitivity and 99% specificity. Further studies researching the combination of SLN detection techniques and PSMA are highly desirable [19]. 

One of the most important strengths of this study is the mapping aspect of the SLN technique. The complexity of the issue lies in the prostate’s lymphatic drainage pathways, which are very diverse and heterogeneous [18]. According to the concept of metastatic spread through the lymphatic vessels, it could be assumed that the absence of tumor invasion in the SLNs indicates that other LNs should be malignancy-free as well [10]. Ideally, performing SLND could prevent overtreatment of the ePLND method in patients with PCa [20]. Currently, PLND performed during RP has been proven to be the most accurate and reliable nodal staging procedure, and therefore it is the gold standard for assessment of nodal involvement in PCa, although therapeutic advantages of LND remain unclear and are being questioned more frequently [18,21,22,23]. Thus, the SLN detection techniques and SLND are investigated to strike a balance between the lower complication burden of SLND and the high staging accuracy of ePLND. Our study is an attempt to assess SLN detection efficacy to eventually decrease the extent of LND and avoid adverse effects of wide-range lymphadenectomy.

What requires further research is the connection between the primary landing site in LNs and the further development of metastasis. We have found that among the SLNs detected scintigraphically, there were 18 in the presacral area, 11 in the common iliac, and only seven in the internal iliac compartment. However, no presacral LNs occurred invaded, and two internal iliac LNs were confirmed to be positive; the first of the latter was not detected pre- or intraoperatively. This indicates that there is a relevant gap between lymphatic drainage pattern variants and actual metastasis-related routes. Moreover, Joniau et al. indicated the importance of the presacral LNs, whereas the results of our study suggest they would preferably be omitted [18]. In addition, during lymphadenectomy, we removed 107 LNs from Marcille’s fossa, 84 from the presacral region and 101 from the common iliac region, and none of those occurred invaded. Radio-guided SLN detection revealed only 34 SLNs in this area. Furthermore, recent randomized clinical trials by Touijer et al. and Lestingi et al. revealed no improvement in oncological outcomes when comparing extended and limited LND templates [6,7]. These studies and the above-mentioned data strongly suggest, despite significant drainage to Marcille’s fossa shown in some studies, that this area, as well as presacral and common iliac regions, may be safely omitted in most patients [12,24,25,26]. 

We acknowledge certain limitations of our study. Firstly, our LND templates are mainly based on the combination of the EAU guidelines and our clinical experience. Therefore, some dominant differences occur in nomenclature, although our broader—in terms of the template extent—approach has helped establish the meaning of some lesser assessed LNs (e.g., presacral nodes). On the other hand, this study is the only investigation of the SLN technique that comprises lymphadenectomy of all pelvic LNs, including Marcille’s fossa. Therefore, we defined this template as ‘modified-extended’ or ‘the true pelvis’. Secondly, the study population is relatively small, and all included patients were admitted to one tertiary center. The low number of patients is the result of our concerns associated with significantly extended LND extent and possible complications. Thus, our findings should be perceived as preliminary. It is worth mentioning that the only observed complications included lymphorrhea and a lymphocele, which are both side effects associated with the LND procedure itself. Therefore, it was difficult to distinguish the specific adverse effects of the SLN detection techniques. Moreover, the SLN detection technique and wide range of lymphadenectomy were non-standard; hence, very few patients agreed to apply these custom procedures. Thirdly, the risk of LNI in all patients was assessed using the Briganti nomogram, but patient selection for this study was based on the EAU risk groups for biochemical recurrence of localized and locally advanced PCa. This is the reason for the wide range of Briganti nomogram results in the study population (2–49%). Thus, it is important to mention that this may be the cause of a relatively high number of N0 patients (n = 33). Due to the great heterogeneity of PCa patients, especially in the intermediate-risk group, future studies should analyze SLN detection efficacy in carefully selected patients (e.g., only the EAU intermediate-risk group and Briganti nomogram >5%). Fourthly, surgeons conducted open or laparoscopic surgeries because available gamma probes were intended for these techniques. Consequently, our study does not explore the SLN detection efficacy in the setting of laparoscopic or robot-assisted surgery.

## 5. Conclusions

In conclusion, radio-guided SLND has a low diagnostic rate and is a poor staging tool. Standard PLND, described as ePLND in the EAU Guidelines, remains the gold standard in nodal metastases assessment in PCa. Our study indicates that lymphatic drainage of the prostate and actual metastasis routes may vary significantly. Further studies on SLN techniques are highly desirable. Plausibly, the appliance of novel imaging modalities, e.g., PSMA-MRI/CT, would hopefully increase the accuracy of SLN detection.

## Figures and Tables

**Figure 1 cancers-14-05012-f001:**
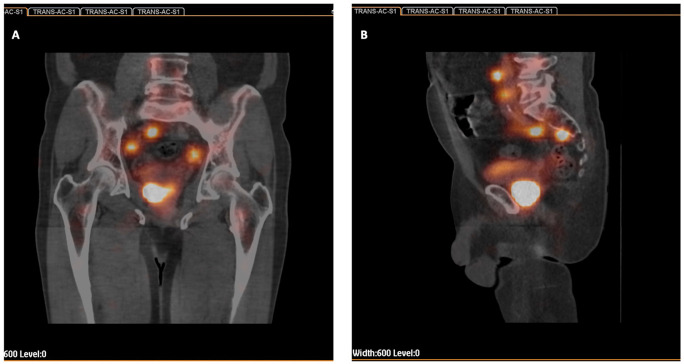
Sentinel lymph node (SLN) procedure in a prostate cancer patient. Fused SPECT-CT images to facilitate anatomic identification of the SLNs: (**A**) Frontal exposition. (**B**) Sagittal exposition.

**Figure 2 cancers-14-05012-f002:**
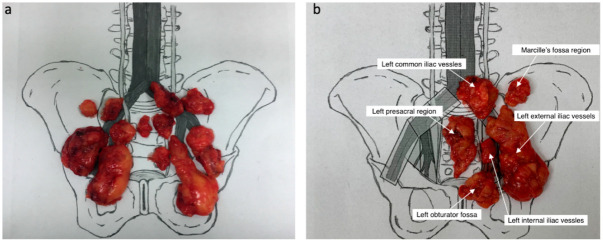
Sentinel lymph node specimens mapped on the scheme (**a**) and grouped into specific lymph node regions (**b**).

**Figure 3 cancers-14-05012-f003:**
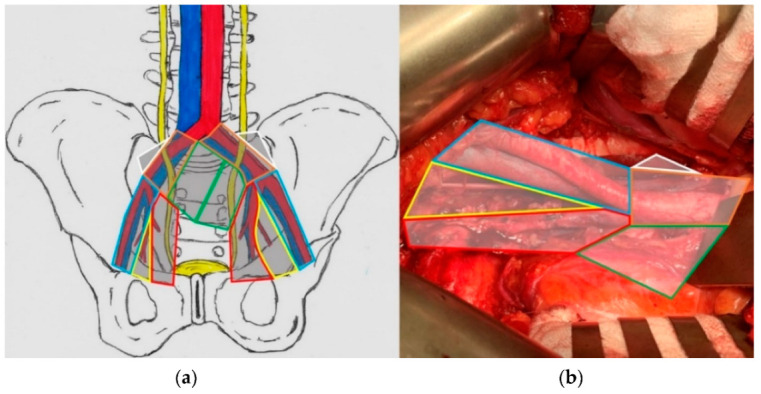
The template of modified-extended lymphadenectomy in the scheme (**a**) and during surgery (**b**); blue—external iliac region, yellow—obturator fossa region, red—internal iliac region, green—presacral region, orange—common iliac region, white—Marcille’s fossa (lying behind the plane).

**Figure 4 cancers-14-05012-f004:**
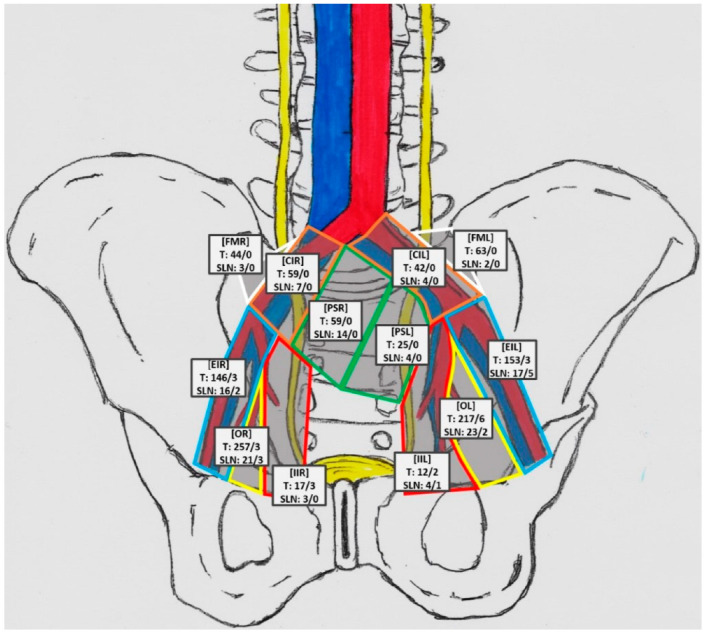
Topography of sentinel nodes in individual anatomical regions, considering the number of LNs removed. Explanation of labels—top line: [region name]; middle line: number of removed LNs/number of LN+; bottom line: number of SLNs/number SLNs+; FMR: Marcille’s fossa right, FML: Marcille’s fossa left, CIL: common iliac left, CIR: common iliac right, PSR: presacral right, PSL: presacral left, EIR: external iliac right, EIL: external iliac left, OR: obturator right, OL: obturator left, IIR: internal iliac right, IIL: internal iliac left.

**Table 1 cancers-14-05012-t001:** Patient demographics and clinical characteristics.

	Patients N0 = 33	Patients N+ = 10	All Patients = 43
Age mean (years); median (range)	64 years; median 66 (50–71)	61.4 years; median 61 (49–70)	63.4 years; median 66 (49–71)
Preoperative PSA mean (ng/mL); median (range)	13 ng/mL; median 12 (2.41–48.2)	16 ng/mL; median 9 (6.28–33.52)	14.02 ng/mL; median 11.65 (2.41–48.2)
Clinical T Stage, n (%)			
cT1	14	4	18 (41.9%)
cT2	16	3	19 (44.1%)
cT3a	2	0	2 (4.6%)
cT3b	1	3	4 (9.4%)
Biopsy Gleason grade group, n (%)			
1 (3 + 3)	10	3	13 (30.2%)
2 (3 + 4)	11	3	14 (32.5%)
3 (4 + 3)	6	1	7 (16.3%)
4 (4 + 4)	3	2	5 (11.6%)
5 (4 + 5 or 5 + 4)	3	1	4 (9.4%)
Pathologic T Score, n (%)			
pT2a	1	0	1 (2.3%)
pT2c	19	0	19 (44.1%)
pT3a	10	5	15 (35.0%)
pT3b	3	5	8 (18.6%)
Pathological Gleason grade group, n (%)			
1 (3 + 3)	4	0	4 (9.4%)
2 (3 + 4)	17	4	21 (48.7%)
3 (4 + 3)	9	4	13 (30.2%)
4 (4 + 4)	0	1	1 (2.3%)
5 (4 + 5 or 5 + 4)	3	1	4 (9.4%)
Briganti nomogram, mean (%); median (range)	12.35%; median 8 (2–48)	22%; median 22 (3–49)	14.36%; median 9 (2–49)
Nodes removed per patient (no.); median (range)	25; median 25 (14–52)	26; median 28 (16–33)	26; median 26 (14–52)

LN: lymph node; PSA: prostate-specific antigen; numbers between parentheses are percentages unless indicated otherwise.

**Table 2 cancers-14-05012-t002:** Diagnostic values of SLN detection techniques.

Diagnostic Test Parameter	Gamma-Probe	SPECT-CT
**sensitivity**	90%	90%
**specificity**	6.06%	6.06%
**PPV**	22.5%	22.5%
**NPV**	66.67%	66.67%
**ACC**	25.58%	25.58%
**FN rate**	10%	10%

SPECT-CT: single-photon emission computed tomography; PPV: positive predictive value; NPV: negative predictive value; ACC: accuracy; FN: false negative.

**Table 3 cancers-14-05012-t003:** Assessment of the correct pN staging depending on the variant of the LND template.

Variant of Lymphadenectomy	N+ Patients Correctly Staged, No. (%)	N+ Patients with All Metastases Removed No. (%)	N+ Lymph Nodes Removed, No. (%)	Lymph Nodes Removed, No.
obturator	4 (40%)	2 (20%)	9 (47.37%)	474
limited lymphadenectomy: obturator + external iliac	9 (90%)	8 (80%)	17 (89.47%)	776
standard lymphadenectomy: obturator + external and internal iliac	10 (100%)	10 (100%)	19 (100%)	805
extended lymphadenectomy: obturator + external and internal iliac + common iliac	10 (100%)	10 (100%)	19 (100%)	906
modified-extended lymphadenectomy: above mentioned + presacral + Marcille’s fossa	10 (100%)	10 (100%)	19 (100%)	1097
sentinel lymph node dissection only	9 (90%)	4 (40%)	13 (68.42%)	118

N+: node positive; No.: number.

**Table 4 cancers-14-05012-t004:** Characteristics of the removed sentinel nodes from individual anatomical areas.

Anatomical Region	No. of Patients	SLN	% of All LN from Station	SLN+ Count	%SLN+/SLN from Station
obturator left (OL)	17	23	10.60%	2	8.70%
obturator right (OR)	16	21	8.17%	3	14.29%
external iliac left (EIL)	14	17	11.64%	5	29.41%
external iliac right (EIR)	12	16	10.26%	2	12.5%
internal iliac left (IIL)	4	4	33.33%	1	25%
internal iliac right (IIR)	3	3	17.65%	0	0%
presacral left (PSL)	3	4	16%	0	0%
presacral right (PSR)	13	14	23.73%	0	0%
fossa of Marcille left (FML)	2	2	3.17%	0	0%
fossa of Marcille right (FMR)	3	3	6.82%	0	0%
common iliac left (CIL)	4	4	9.52%	0	0%
common iliac right (CIR)	5	7	11.86%	0	0%

No.: number; SLN: sentinel lymph nodes; SLN+: positive sentinel lymph nodes.

## Data Availability

The data is available from the Authors upon reasonable request.

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
