# Peer review of "Diagnostic Value of Radio-Guided Sentinel Node Detection in Patients with Prostate Cancer Undergoing Radical Prostatectomy with Modified-Extended Lymphadenectomy"

_cancers, 2022, doi:10.3390/cancers14205012_

Round 1

Reviewer 1 Report

Good paper, not many look at Spect- CT, but think about use of Choline PET or PSMA PET in the Salvage situation, these are far more senstive and specific. 

Author Response

 Good paper, not many look at Spect- CT, but think about use of Choline PET or PSMA PET in the Salvage situation, these are far more senstive and specific.

Response: Thank you very much for this encouraging and valuable comment. Indeed, the use of SPECT-CT is limited. We are currently working on a study including PET-CT PSMA in the sentinel node technique in patients qualified for prostatectomy with lymphadenectomy. Radioguided surgery in salvage settings is also of interest to us.

Reviewer 2 Report

Discussion:

1) some….  studies: please add appropriated references

Reviewer 3 Report

It is an original and detailed article, and the manuscript is well written. Conclusions were made appropriately and had high standards. This manuscript met the goal of Cancers.

Reviewer 4 Report

The manuscript entitled “Diagnostic Value of Radio-Guided Sentinel Node Detection in Patients with Prostate Cancer Undergoing Radical Prostatectomy with Modified-Extended Lymphadenectomy” is interesting and bring new information to prostate cancer approach. Please, see my specific comments below:

1.     The figures captions needs improvement and more detailed description.

2.     The figure 2 was cur in the middle.

3.     Authors could describes better the morbidity associated with lymphadenectomy. Authors described lymphorrhea as the most common side effect and this phrase suggested that other happened. Please, provide more details.

4.     Any association was investigated regarding the presence of metastasis in lymph node and other patient’s factors?   

5.     Authors could discuss better the results, including the side effects of the procedure and if authors indicate or not.
